# Variant Enrichment Analysis to Explore Pathways Disruption in a Necropsy Series of Asbestos-Exposed Shipyard Workers

**DOI:** 10.3390/ijms232113628

**Published:** 2022-11-07

**Authors:** Sergio Crovella, Ronald Rodrigues Moura, Lucas Brandão, Francesca Vita, Manuela Schneider, Fabrizio Zanconati, Luigi Finotto, Paola Zacchi, Giuliano Zabucchi, Violetta Borelli

**Affiliations:** 1Biological Sciences Program, Department of Biological and Environmental Sciences, College of Arts and Sciences, University of Qatar, Doha 2713, Qatar; 2Institute for Maternal and Child Health—IRCCS “Burlo Garofolo”, 34137 Trieste, Italy; 3Department of Pathology, Center of Medical Sciences, Federal University of Pernambuco, Recife 50670-901, Brazil; 4UCO Anatomia e Istologia Patologica, Azienda Sanitaria Universitaria Giuliano Isontina (ASUGI), Hospital of Cattinara, 34149 Trieste, Italy; 5Occupational Health Unit, Local Health Authority, 34074 Monfalcone, Italy; 6Department of Life Sciences, University of Trieste, 34127 Trieste, Italy

**Keywords:** asbestos, mesothelioma, lung cancer, whole exome, FFPE (fixed in formalin and embedded in paraffin), variant enrichment analysis (VEA)

## Abstract

The variant enrichment analysis (VEA), a recently developed bioinformatic workflow, has been shown to be a valuable tool for whole-exome sequencing data analysis, allowing finding differences between the number of genetic variants in a given pathway compared to a reference dataset. In a previous study, using VEA, we identified different pathway signatures associated with the development of pulmonary toxicities in mesothelioma patients treated with radical hemithoracic radiation therapy. Here, we used VEA to discover novel pathways altered in individuals exposed to asbestos who developed or not asbestos-related diseases (lung cancer or mesothelioma). A population-based autopsy study was designed in which asbestos exposure was evaluated and quantitated by investigating objective signs of exposure. We selected patients with similar exposure to asbestos. Formalin-fixed paraffin-embedded (FFPE) tissues were used as a source of DNA and whole-exome sequencing analysis was performed, running VEA to identify potentially disrupted pathways in individuals who developed thoracic cancers induced by asbestos exposure. By using VEA analysis, we confirmed the involvement of pathways considered as the main culprits for asbestos-induced carcinogenesis: oxidative stress and chromosome instability. Furthermore, we identified protective genetic assets preserving genome stability and susceptibility assets predisposing to a worst outcome.

## 1. Introduction

Malignant pleural mesothelioma (MPM) and lung cancer (LC) are diseases associated with asbestos exposure [1,2,3]. However, exposure to this fibrous mineral is not the only cause responsible for the development of these debilitating diseases [4,5], and the mechanisms underlying lung and pleural cell injury have not, as yet, been fully understood. In the last decade, many studies have demonstrated the contribution of genetic factors in promoting asbestos-related carcinogenesis and lung genotoxicity.

A number of studies have compared past asbestos exposure and genetic polymorphisms using either candidate genes approaches [6,7,8,9,10,11,12,13,14,15,16,17,18,19,20,21,22] or whole-genome association studies on both MPM and LC [23,24,25,26].

Despite using separate and well-characterized cohorts of controls and mesothelioma patients, a number of GWAS studies failed to identify common genetic risk factors associated with mesothelioma [19,24,25,27]. Germline mutations in BRCA1-associated protein 1 (BAP1) and some DNA repair genes have been considered as predisposing genetic factors associated with mesothelioma development [11,25,28,29]. However, these genetic risk factors are not specific for mesothelioma and can predispose individuals to other cancers such as uveal melanoma [29].

Furthermore, the impact of individual genetic variations on asbestos-related lung cancer remains partially understood. Candidate gene approach studies demonstrated that polymorphisms in genes encoding for xenobiotic metabolizing enzymes such as glutathione S-transferase M1 (*GSTM*), glutathione S-transferase theta 1 (*GSTT1*), myeloperoxidase (*MPO*), cytochrome P450 family 1 subfamily A member 1 (*CYP1A1*) and cytochrome P450 family 2 subfamily E member 1 (*CYP2E1*), and manganese superoxide dismutase (*SOD2*) are associated with asbestos-related lung cancer risk [9,15]. Although recent genome-wide association studies (GWAS) identified novel loci for lung cancer risk, few investigators have addressed genome–environment interactions. Wei and colleagues suggested that immune-function-regulation-related pathways (Fas pathway) might be mechanistically involved in asbestos-associated lung cancer risk [23]. Liu and co-workers proposed MIRLET7BHG (MicroRNA *Let-7b* Host Gene) as a possible important predictive marker for asbestos-exposure-related lung cancer [30]. Finally, Kettunen and colleagues identified novel DNA methylation changes associated with lung tumors and asbestos exposure [26].

Despite having achieved the identification of novel genetic variants associated with specific pathological conditions, none of these genomic studies were completely successful in unravelling the pathogenic mechanisms at the basis of the diseases. In fact, genetic variants were not integrated in a broader biological context, but considered individually in a unique clinical context [31]. Basing a comprehensive explanation of a disorder only on association studies seeking common variants is likely to fail because of the high statistical pressure that occurs when performing a GWAS. Thus, new approaches should be brought into play to gain an in-depth understanding of the disease complexity. The variant enrichment analysis (VEA) workflow was recently developed [31] as a tool applicable for whole-exome sequencing data, able to find differences between the numbers of genetic variants in a given pathway compared with a reference dataset. GWAS are valuable tools for the identification of genetic variants associated with a certain phenotype, but require a large number of subjects to overcome the statistical pressure. On the other hand, VEA aims to enrich pathways based on genetic variant information. Thus, VEA does not lose variants with small impact on phenotype and does not require a large number of individuals [31]. In a previous study, we identified different pathway signatures associated with the development of pulmonary toxicity in mesothelioma patients treated with radical hemithoracic radiation therapy, using VEA. This allowed us to formulate some hypotheses on the protection from side effects derived from radiotherapy and on factors predisposing to a worst response to the treatment [32].

In the present study, we used VEA to discover novel pathways that were altered in individuals exposed to asbestos who developed, or did not develop, lung cancer or mesothelioma. In the population-based autopsy study we designed, asbestos exposure, the assessment of which constituted a relevant source of heterogeneity in previous genetic studies [13,21], was evaluated and quantitated by investigating objective signs of exposure [33].

## 2. Results

### 2.1. Study and Populations Characteristics

The study was performed on 14 autopsy samples derived from individuals exposed to asbestos who developed either pleural mesothelioma (MPM group, *n* = 7: epithelial histological type *n* = 2, biphasic histological type *n* = 2 and sarcomatoid histological *n* = 3) or lung cancer (LC group, *n* = 7: small-cell LC *n* = 2 and non-small-cell LC *n* = 5) using, as a reference, 5 autopsy samples from asbestos-exposed individuals who died of asbestos-unrelated diseases (CTRL group, *n* = 4). Patients’ characteristics are summarized in Table 1.

All individuals were male with a mean age of 79.4 ± 2.2 (CTRL); 76.1 ± 8.6 (LC) and 76.9 ± 6.3 (MPM) years at the time of death. Smoking habit was available only for the LC groups, in which all individuals were smokers (two current and two former).

All individuals were employed in the shipyard and displayed similar asbestos exposure due to working condition, which was validated by the presence of pleural plaques and characterized qualitatively (type of asbestos) and quantitatively by the analysis of asbestos (bodies and/or fibers: ABs and AFs, respectively) lung burden. Pleural plaques of grade 2 were mostly present in all cases, except one of grade 1 in the MPM group.

ABs lung burden was, for all cases, well above the threshold for occupational exposure (10^3^ AB/g) [34], ranging from 0.62 × 10^5^ to 1.98 × 10^5^ AB/g of dry tissue (mean ± sd: 1.30 × 10^5^ ± 6.07 × 10^4^) in the CTRL group, from 0.064 to 4.300 × 10^5^ AB/g of dry tissue (mean ± sd: 1.02 × 10^5^ ± 1.48 × 10^5^) in the LC group, and from 0.27 × 10^5^ to 2.91 × 10^5^ from AB/g of dry tissue (mean ± sd: 1.27 × 10^5^ ± 9.9 × 10^4^) in the MPM group.

AF (length >1μm) lung burden was determined for all LC and MPM cases and was far above the threshold for occupational exposure (>1 million/g) [35] in all cases (except one, in which was 0.57 × 10^6^/g), ranging from 0.57 × 10^6^ to 40 × 10^6^ AF/g of dry tissue (mean ± sd: 1.04 × 10^7^ ± 1.37 × 10^7^) in the LC group and from 1.5 × 10^6^ to 73 × 10^6^ AF/g of dry tissue (mean ± sd: 1.86 × 10^7^ ± 2.72 × 10^7^) in the MPM group. Amphibole asbestos exposure was confirmed (>80%) by mineralogical analysis in all cases of LC and MPM.

The small number of analyzed subjects is due to the restrictive enrolment criteria applied in order to possess a population study of individuals with similar exposure to asbestos who did or did not develop asbestos-related cancers.

### 2.2. Genetic Analysis

The WES data obtained from CTRL, LC and MPM patients, including genetic variation information, are detailed as Appendix A).

When we searched our dataset for variants described in previously published GWAS on MPM and asbestos exposure, we found none were present in our WES data, reinforcing our belief that a simple genetic variant analysis cannot explain the etiopathological mechanisms underlying multifactorial phenotypes.

Here, we describe our WES analysis approach focusing on VEA findings for the three groups. All the information on the VEA pipeline, as well as the statistical methods, is reported in the methods section of this article. Firstly, we were able to determine which pathways carry a higher number of genetic variants compared to the general population by using the false discovery rate (FDR) method. Then, based on a Venn diagram, we identified exclusive and shared enriched pathways among the studied groups (Figure 1).

### 2.3. Exclusive Enriched Pathways (EeEP)

One pathway was identified only in the CTRL group: Nef-mediated downregulation of MHC class I complex cell surface expression (R-HSA-164940) (Table 2).

Five pathways are exclusive to the LC groups (Table 2):A tetrasaccharide linker sequence is required for glycosaminoglycans (GAG) synthesis (R-HSA-1971475);HS-GAG degradation (R-HSA-2024096);Defective beta-1,4-Galactosyltransferase 7 (B4GALT7) causes Ehlers–Danlos syndrome (EDS), progeroid type (R-HSA-3560783);Defective galactosylgalactosylxylosylprotein 3-beta-glucuronosyltransferases 3 (B3GAT3) causes joint dislocations, short stature, craniofacial dysmorphism, and congenital heart defects (JDSSDHD) (R-HSA-3560801);Defective -beta-1,3-galactosyltransferase 6 (B3GALT6) causes EDSP2 and spondyloepimetaphyseal dysplasia with joint laxity type 1 (SEMDJL1) (R-HSA-4420332).

Eleven pathways were found to be exclusively enriched in the MPM group (Table 2):Beta Klotho-mediated ligand binding (R-HSA-1307965);Degradation of the extracellular matrix (R-HSA-1474228);Scavenging of heme from plasma (R-HSA-2168880)Nuclear envelope (NE) reassembly (R-HSA-2995410);Laminin interactions (R-HSA-3000157);Non-integrin membrane–ECM interactions (R-HSA-3000171);Defective Solute Carrier Family 29 Member 3 (SLC29A3) causes histiocytosis-lymphadenopathy plus syndrome (HLAS) (R-HSA-5619063);Infectious disease (R-HSA-5663205);Keratinization (R-HSA-6805567);Runt-related transcription factor 2 (RUNX2) regulates bone development (R-HSA-8941326);Interferon alpha/beta signaling (R-HSA-909733).

### 2.4. Shared Enriched Pathways (SeEP)

Two pathways are shared by the CTRL and the MPM groups: immunoregulatory interactions between a lymphoid and a non-lymphoid cell (R-HSA-198933) and the DNAX-activating protein of 12 kDa (DAP12) interactions (R-HSA-2172127).

Five pathways are shared among the CTRL, LC and MPM groups: Defective Polypeptide *N*-Acetylgalactosaminyltransferase 12 (GALNT3) causes familial hyperphosphatemic tumoral calcinosis (HFTC) (R-HSA-5083625); Defective C1GALT1 Specific Chaperone 1 (C1GALT1C1) causes Tn polyagglutination syndrome (TNPS) (R-HSA-5083632); Defective GALNT12 causes colorectal cancer 1 (CRCS1) (R-HSA-5083636); Dectin-2 family (R-HSA-5621480); and Termination of O-glycan biosynthesis (R-HSA-977068). These pathways are not involved in the neoplastic transformation and, as they do not contribute to the determination of the susceptibility to develop asbestos-related cancers, they will not be further considered in this manuscript.

Finally, four pathways are shared by the LC and the MPM groups: Fibroblast Growth Factor Receptor 4 (FGFR4) mutant receptor activation (R-HSA-1839128); Defective EXT2 causes exostoses 2 (R-HSA-3656253); Defective EXT1 causes exostoses 1, Trichorhinophalangeal syndrome type II (TRPS2) and CHDS (R-HSA-3656253); and Retinoid metabolism and transport (R-HSA-975634).

## 3. Discussion

Post-mortem FFPE tissue samples are an important resource for uncovering a potential genetic signature to predict susceptibility to asbestos-related cancers. FFPE allows for the selection of individuals who are well-characterized in terms of clinical–pathological presentation and exposure features, thereby excluding the source of heterogeneity of previous genetic studies [13,21]. Historically, FFPE samples were not considered a viable source for molecular analysis since nucleic acids can be significantly modified by protein–nucleic acid and protein–protein cross linking. However, in recent years, the methods and protocols for extracting nucleic acids and proteins from FFPE tissue specimens have significantly improved [36], and WES may now be assayed using DNA isolated from archival FFPE samples.

In this study, restrictive enrolment criteria were applied to obtain a necropsy population of individuals with similar exposure to asbestos who did or did not develop asbestos-related cancers. Although the application of these criteria reduced the number of enrolled individuals, this strict selection of exposed individuals represents one of the strong points of our study. This workflow, which includes using FFPE tissues as a source of DNA, WES analysis and VEA to interpret WES findings were recently developed [31] to study small numbers of individuals with the aim of increasing the amount of biologically useful information retrievable from WES. VEA can identify disrupted pathways in several individuals with the same phenotype. The ability to translate exome variant data into predictions of pathway alterations can assist in mechanically deciphering the pathogenic mechanisms of the asbestos-related diseases.

### 3.1. VEA Applied to Thoracic Cancers Induced by Asbestos Exposure

Inhalation of asbestos fibers represents the causative agent of pulmonary diseases such as benign pleural fibrosis, plaques and asbestosis, lung cancer and malignant mesothelioma. Several mechanisms have been identified as responsible for both benign and malignant outcomes of asbestos exposure. They can be as diverse as genomic alteration, intracellular signaling cascade activation, generation of reactive oxygen and nitrogen species, as well as direct mechanical cell damage [37]. While risk factors are undoubtedly involved in the development of asbestos-related cancers, it is still difficult to predict which exposed individual will develop a malignant disease and which will not.

In our study, we identified potentially disrupted pathways in individuals who either developed or did not develop thoracic cancers induced by asbestos exposure. VEA analysis allowed us to confirm previously hypothesized mechanisms of asbestos carcinogenesis and to identify novel molecular pathways characterizing different outcomes of asbestos exposure.

In Figure 1, we describe the pathways, as determined by VEA, involved in the three outcomes of asbestos exposure: no asbestos-related cancers (CTRL), lung cancer (LC) or malignant pleural mesothelioma (MPM).

#### 3.1.1. Exclusive Enriched Pathways (eEP) for Controls (CTRL)

The exclusive enriched pathway (eEP) found in CTRL individuals was related to “Nef mediated downregulation of MHC class I complex cell surface expression” (R-HSA-164940). Nef protein function is dependent on its interaction with the phosphofurin acidic cluster sorting (PACS) protein PACS-1 [38,39]. Mutation or altered expression of PACS proteins are associated with conditions ranging from cancer, obesity and viral pathogenesis to epilepsy and neurodegenerative disorders [40]; however, the regulation mechanism of PACS-1 is still unknown. Recent reports have suggested a role for PACS-1 in maintaining chromosomal integrity [41], with PACS-1 overexpression resulting in genomic instability of human cancer cells [42]. The (PACS-1)-dependent protein-sorting pathway, with its implication in genomic instability, could represent a novel crucial event in asbestos-related cancer development since its impairment seems to be exclusive to asbestos-exposed individuals who did not develop LC or MPM (Figure 2).

#### 3.1.2. Exclusive Enriched Pathways (eEP) for LC

All five of the defective eEPs in the LC group are involved in heparan sulphate glycosaminoglycans (HS-GAGs)/proteoglycans (HSPGs) biosynthesis and degradation (Figure 2, point 1).

R-HSA-3560801 [defective galactosylgalactosylxylosylprotein 3-beta-glucuronosyltransferases 3 (B3GAT3)], R-HSA-4420332 [defective beta-1,3-galactosyltransferase 6 (B3GALT6)] and R-HSA-1971475 are involved in the formation of a tetrasaccharide linker sequence required for glycosaminoglycans (GAGs) biosynthesis, while R-HSA-3560783 (defective B4GALT7 galactosyltransferase) is essential in proteoglycan synthesis. On the other hand, R-HSA-2024096 is involved in HS-GAG degradation [43].

Heparan sulfate proteoglycans (HSPGs) are widely distributed in mammalian tissues. They are involved in several processes related to malignancy and, it has emerged, play key roles in tumor initiation and progression [44]. Deregulation of HSPGs resulting in malignancy may be due to either their abnormal expression levels or to changes in their structure and functions, as a result of the altered activity of their biosynthetic or remodeling enzymes [45]. Interestingly, Human sulfatase-2 (SULF2), an HS 6-O-endosulfatase involved in modulating HS biological activities, and B4GALT7 have been previously found to promote, respectively, carcinogenesis [46] and progression [47] in lung cancers. Furthermore, two enriched exclusive pathways (eEP) shared by LC and MPM individuals were related to Heparan sulfate (HS) synthesis (R-HSA-3656237 and R-HSA-3656253), suggesting the possible involvement of HSPGs dysregulation in asbestos-related cancer pathogenesis (Figure 2, point ➊).

#### 3.1.3. Exclusive Enriched Pathways (eEP) for MPM

##### Epithelial–Mesenchymal Transition (EMT) as the First Step in MPM Tumorigenesis (Figure 2, Point ➋)

Among the defective eEPs in the MPM group, R-HSA-1307965 is involved in FGF19-FGFR4 signaling. Interestingly, an eEP common to both LC and MPM is also related to FGFR4 receptor activity (R-HSA-1839128).

The FGF19 subfamily (hormone-like FGFs) binds to FGFR and its coreceptor, Klotho, to drive endocrine signaling [48]. β-Klotho forms a complex with FGFR4, enhancing the affinity between FGF19 and FGFR4 [49]. As previously described, the FGFR4 receptor activation pathway (R-HSA-1839128) is defective in both LC and MPM, confirming the possible involvement of FGF19-FGFR4 signaling in the development of asbestos-related cancers.

Among the main signaling pathways downstream of FGF19-FGFR4 interaction, epithelial-to-mesenchymal transition (EMT) [50,51,52] could be relevant in asbestos-induced mesothelioma.

EMT is a process through which epithelial cells acquire mesenchymal features [53,54,55]. Morphological and molecular alterations induced by crocidolite and chrysotile asbestos are suggestive of EMT [56,57], which may serve as the initial step in MPM tumorigenesis [58] transforming adherent cells in motile cells able to migrate and invade the extracellular matrix [54].

Interestingly, among the other defective pathways in the MPM group, one (R-HSA-8941326) affects *RUNX2*, the master gene in osteogenesis, which has been shown to increase expression of *EMT* genes (included in R-HSA-8941326) vimentin, *TWIST1* and *SNAIL1* in lung adenocarcinoma cells [59].

##### The ECM Shaping of a Pro-Tumor Microenvironment (Figure 2, Point ➊)

An exclusive pathway found in MPM individuals was one related to the degradation of the extracellular matrix by matrix metalloproteases (MMPs). MMPs play an important role in lung remodeling in response to environmental agents and represent the most prominent family of proteinases associated with tumorigenesis [60].

Cell invasiveness is one of the hallmarks of cancer, and mesothelial cells acquire the ability to invade the matrix upon transformation. This process has been shown to be linked to the enhanced secretion of metalloproteinase, mainly MMP-2 and MMP-9, both of which are capable of acting as EMT inducers [61,62]. MMP-2 secretion from human normal mesothelial MeT-5A cells has been shown to increase upon treatment with chrysotile and to induce EMT [57], suggesting that changes in the surrounding microenvironment render the ECM more amenable to degradation and invasion [57,62,63] (Figure 2, point ➋). Certainly, the underlying mechanism of MMP-2-induced EMT in MMP development deserves further study.

A potential gene–environment interaction between MMP SNPs and asbestos, which is the major risk factor for MMP, has been previously shown, with certain genetic variants in MMP genes exerting either protective or tumor-promoting effects on mesothelioma development [64]: (i) an Italian-based GWAS study found that *MMP14*, among other genes, could be a risk factor for MPM [25] and (ii) MMP2 rs243865 polymorphism has been described as protective for pleural mesothelioma development [65]. R-HSA-1474228 disruption in MPM individuals is a further confirmation that MMP polymorphisms influence the risk of mesothelioma.

Pleural mesothelial cells are involved in the production of the sub-mesothelial connective tissue matrix of pleura and lung [66]. In normal pleura, fibronectin and laminin are components of the basement membrane [66,67] and these ECM proteins can modulate the adhesion and proliferation of mesothelial cells [68], playing a key role in establishing a fertile soil for individual tumor cells to originate primary and secondary tumors [69].

Matrix and matrix-associated components (laminins (R-HSA-3000157) and proteoglycans (R-HSA-3000171)) are collectively regulated by epithelial or mesothelial cells, fibroblasts and resident immune cells to orchestrate tumor dormancy or outgrowth in the lung and pleura, respectively. The identification of two disrupted pathways involving ECM-associated components in MPM individuals suggests they may play a role in establishing dormancy or outgrowth of asbestos-transformed cells (Figure 2, point ➊).

##### Asbestos Fibers and the Nuclear Envelope (Figure 2, Point ➌)

Reassembly of the nuclear envelope (NE) around separated sister chromatids begins in late anaphase and is completed in the telophase [70]. Nuclear envelope–fiber attachment has been shown in asbestos-exposed mesothelial cells during telophase, when a chromatin strand ran with the fiber into the intercellular bridge [71]. Such strands may break, causing chromosome structural rearrangements [72]. The disruption of the nuclear envelope reassembly pathway found in MPM individuals could foster the asbestos-induced genomic changes which are considered among the initial events leading to MPM (Figure 2, point ➌).

##### The Emerging Role of Free Heme in MPM Pathogenesis (Figure 2, Point ➍)

Chrysotile asbestos fibers are potent inducers of hemolysis [73,74,75,76] that results in hemoglobin release [73]. Free hemoglobin in plasma is scavenged by the extracellular protein haptoglobin, but when the buffering capacity of plasma haptoglobin is overwhelmed, free heme is released [77]. Non-encapsulated/free heme can cause tissue damage since it intercalates with biologic membranes and causes oxidative stress by generating reactive oxygen species (ROS) and redox-active iron capable of initiating lipid peroxidation [78], and has been shown to be an important injurious agent for the lung following Libby amphibole asbestos exposure [79]. Efficient Heme scavenging has been shown to reduce pulmonary endoplasmic reticulum stress, fibrosis, and emphysema [80].

Interestingly, asbestos exposure induces higher serum levels of haptoglobin, and a genetic polymorphism of haptoglobin, the phenotype Hp1-1, has been found more frequently in exposed individuals who developed asbestosis [81]. In addition, haptoglobin has recently been included in serum exosomal proteomic signatures relevant to asbestos exposure potentially validated as candidate biomarkers [82]. The disruption of heme scavenging from plasma (R-HSA-2168880) could then induce endoplasmic reticulum/oxidative stress, thereby increasing the risk of developing mesothelioma (Figure 2, point 4). Furthermore, the risk of developing mesothelioma has previously been associated with Heme oxygenase (HO)-1, a rate-limiting enzyme of heme degradation which plays a protective role against oxidative stress: Murakami and colleagues suggested that long (GT)n repeats in the HO-1 gene promoter are associated with a higher risk of malignant mesothelioma in the Japanese population [ 17].

#### 3.1.4. The Possible Involvement of Retinol Metabolism and Transport and the Susceptibility to Develop MPM and LC (Figure 2, Point ➎)

The possible involvement of retinol metabolism (retinoid metabolism and transport (R-HSA-975634)) in the pathogenesis of both asbestos-induced thoracic cancers, emerging for the first time from our VEA analysis, could contribute to explain the failure of the Beta-Carotene and Retinol Efficacy Trial (CARET) for the prevention of asbestos-related cancers. CARET was a population-based cancer prevention program providing retinol supplements to individuals who were at high risk for lung cancer because of a history of smoking or asbestos exposure [83,84]. The program was stopped ahead of schedule due to increased incidence of lung cancer among the participants [85,86]. The retinoid metabolism and transport pathway disruption we found in exposed individuals who developed LC or MPM could support a possible role for retinol/vitamin A metabolism in asbestos-related cancer pathogenesis (Figure 2, point ➎).

### 3.2. Strength and Limitation of the Study

The evaluation of asbestos exposure based on objective signs of exposure excluded a relevant source of heterogeneity of previous genetic studies but, at the same time, limited the number of subjects included in the analysis. Our VEA approach, however, can be applied in the case of individual reports or in studies with low numbers of subjects, where the statistical power of genetic findings, such as in the case of GWAS, fails.

## 4. Materials and Methods

### 4.1. Sample Collection and DNA Extraction

Tissue samples were obtained from a necropsy series (1983–2015) of asbestos-exposed shipyard workers with asbestos-related neoplastic diseases (malignant pleural mesothelioma (MPM) and lung cancer (LC)) and from a control population (CTRL) as described previously by Crovella and colleagues [20,87]. The CTRL population was composed of shipyard workers exposed to asbestos, who did not develop asbestos-related tumors, neither MPM or lung cancer LC nor other types of asbestos-induced tumors, such as laryngeal, gastrointestinal and ovarian cancer [35], and died of other causes after 75 years of age.

All tissue samples originated from the Monfalcone area (Northeastern Italy) as described previously by Crovella and colleagues [20]. Monfalcone is a small industrial town with large shipyards. This area is characterized by a very high incidence of asbestos-related mesothelioma [88] and by a very high prevalence of pleural plaques in the necropsy population [89].

Asbestos exposure was objectively established for all autopsies by evaluation, during the necroscopic examination, of the presence of pleural plaques and/or asbestos (bodies and fibers) lung burden [90]. Population/sample characteristics are summarized in Table 1.

Asbestos body (AB) and asbestos fibers (AF) per gram of dry lung tissue were counted using optical [91,92] and scanning electron microscopy, respectively [93], and expressed as geometric means and standard deviations.

Pleural plaques were examined and classified in three stages, as previously described [20,87].

Diagnosis of pleural mesothelioma or lung cancer was confirmed or obtained during necropsy and assessed by histological examination. Information on histological type of neoplasia, both mesothelioma and lung cancer, was collected for each patient. The malignant pleural mesothelioma cases were differentiated into three different histological types: (1) epithelioid, possessing mostly cells with epithelial morphology; (2) sarcomatoid, with cells having a spindle morphology; (3) biphasic, with cells belonging to both categories (spindle and epithelioid) [94].

Lung cancers were classified into two large histological categories: small-cell lung cancer (SCLC, *n* = 2) and non-small-cell lung cancer (NSCLC, *n* = 5) [95]. All LC patients were smokers.

The study was approved by the regional ethics committee of Friuli Venezia Giulia (Parere CEUR-2020-PR-11—seduta dd 04/08/2020—odg 4.1).

The archives of the Department of Pathological Anatomy of the Hospital of Monfalcone stored the histological samples from all autopsies. Myocardial tissue was chosen as the starting material for DNA extraction, being free from neoplastic cells and thus without somatic alterations due to tumorigenic transformation [96]. Mean age (years) ± sd (se) of the material at the time of DNA extraction was 33.4 ± 4.7 (2.1) (controls) and 9.1 ± 2.1 (0.6) (study) years, ranging from 6 to 38 years. Fixation was made in 10% formalin for all the samples; from the same paraffin block, 40–50 slices were cut with a 5–7 μm thickness and processed for DNA extraction.

### 4.2. FFPE DNA Extraction

High-quality genomic DNA is a critical step for attaining high-quality results in next-generation studies, so we compared two commercial kits, the QIAamp DNA FFPE Tissue Kit (Qiagen, Hilden, Germany) and the ReliaPrep™ FFPE gDNA Miniprep (Promega, Madison, WI, USA) examining the yield and quality of DNA extracted from the same FFPE tissue: the Promega kit performed better than the Qiagen one and was therefore used in all experimental settings. Moreover, in order to improve the quality of the DNA extracted from our samples, we employed a xylene-based protocol for removing paraffin, as follows: three washes of the sections with xylene at 50 °C, followed by three washes in graded ethanol (from 100% to 50%) to remove xylene, and final resuspension in the Promega kit lysis buffer. Having eliminated paraffin contamination with this protocol, still the quantity of extracted DNA was not sufficient for NGS. We, therefore, increased the starting material (from 40–50 slice to 80–100) and incubation with Proteinase K from 1 h to overnight at 37 °C. Using this modified protocol, we were able to obtain enough yield of genomic DNA with a quality suitable for NGS.

Absorbance at 260 nm, 280 nm and 230 nm was then measured using a Nanodrop 2000 (ThermoFisher, Waltham, MA, USA) spectrophotometer, to assess the quantity and purity of the DNA extracted from FFPE. The level of DNA degradation was then assessed qualitatively using agarose gel electrophoresis and quantitatively by ProNex^®^ DNA QC Assay (Promega) following the manufacturer’s instructions. After all experimental procedures and adjustments to the protocols as described above, the quality of the DNA was acceptable for NGS sequencing as stated by our NGS service provider Macrogen. Finally, in the NGS analysis, even if we took all the precautions to avoid artifacts due to sample fixation, we paid attention to the fact that formalin treatment could result in artificial C > T or G > A mutations.

### 4.3. Exome Sequencing

DNA extracted following the protocol described above was dispatched to Macrogen Europe (Amsterdam, The Netherlands) for next-generation sequencing. Following a subsequent quality check control, sample DNA was fragmented and enriched using the pair-ended Exome library (SureSelect V6 Post Agilent^®^) to obtain a library of coding DNA. Then, the sample was sequenced using Illumina^®^ NovaSeq S4 300 to obtain roughly 12 Gb of raw data for each sample.

Raw data obtained from sequencing were firstly processed for quality control using the application fastQC (https://www.bioinformatics.babraham.ac.uk/projects/fastqc, accessed on 20 November 2021), in which an overall summary of the sequencing performance can be assessed (e.g., total sequences, sequence length, GC proportions, sequence quality score, and adapter content). After that, library adapters and reads with lengths below 25 base pairs and with low Phred score (Q < 20) were removed using the TrimGalore application (http://www.bioinformatics.babraham.ac.uk/projects/trim_galore/, accessed on 20 November 2021). Unmapped reads were aligned to the hg38 reference genome using the BWA algorithm [97], and then soft-clipped and duplicated reads removal as well as base quality score re-calibration was performed using Picard Tools v. 2.7.0 and GATK v. 4.1.2.0, respectively (https://broadinstitute.github.io/picard/ (accessed on 20 November 2021) and https://software.broadinstitute.org/gatk/ (accessed on 20 November 2021)). For variant calling, Strelka2 software was employed [98]. WES annotation for each sample was performed using Annovar software [99]. Although filtering out duplicated reads and soft-clipped bases reduces dramatically the occurrence of miscalled variants due to FFPE artefacts, we set the following filters for variant calling: base quality (GQ) > 30 and alternate allele depth (AD) > 10.

Variant enrichment analysis (VEA), described in detail elsewhere [31], was applied using the Non-Finnish European (nfe) dataset from GnomAD v3.4 [100]. In summary, VEA evaluates if the number of generic variants found in an individual (or group of individuals) presented in a reactome pathway [43] is statistically different from the number of variants presented in the same pathway found in a reference dataset (in this case, the nfe population). For group data, we used Venn diagrams to summarize “enriched” pathways exclusive for each group.

## 5. Conclusions

In this study, we used a WES-based VEA approach to identify potentially disrupted pathways in individuals who developed, or did not develop, thoracic cancers induced by asbestos exposure. Here, we describe these molecular pathways whose relevance is also supported by the literature, and formulate a hypothesis on how the dynamic crosstalk between asbestos exposure and germline genetic asset (susceptibility) can have an impact on asbestos-related thoracic cancer pathogenic mechanisms.

By using VEA analysis, we confirmed the involvement of the pathways considered as the main responsible for asbestos-induced carcinogenesis: oxidative stress (heme) and chromosome instability. Furthermore, we identified novel molecular pathways associated with different outcomes of asbestos exposure: a protective role of PACS-1 in maintaining genome stability, and factors, such as the dysregulation of ECM dynamics and the induction of EMT transition, predisposing to a worst outcome (LC and MPM).

Susceptibility of asbestos-induced LC seems to be limited to dysregulation of ECM dynamics (and possibly to that of FGF19-FGFR4-induced EMT) while susceptibility to MPM seems more cancer-specific, retracing various aspects that have long been considered as key factors of asbestos-induced carcinogenesis: oxidative stress (with the novelty of heme involvement), chromosomal aberration due to nuclear–asbestos interaction, and EMT that both probably reflect the peculiarity of mesothelial cell cytoskeleton. As far as the common aspects of LC and MPM are concerned, our data could represent a starting point for future investigation on the central role of ECM in asbestos-induced pathogenesis, since the pathways involved in ECM dynamics (composition, synthesis, and degradation) appear to be hallmarks of both these asbestos-induced thoracic cancers.

In addition, our WES-based VEA approach identified defective retinoid metabolism and transport as possible risk factors for developing asbestos-induced thoracic cancers, supporting data in the literature on the failure of asbestos-related cancer prevention programs based on retinol supplementation.

## Figures and Tables

**Figure 1 ijms-23-13628-f001:**
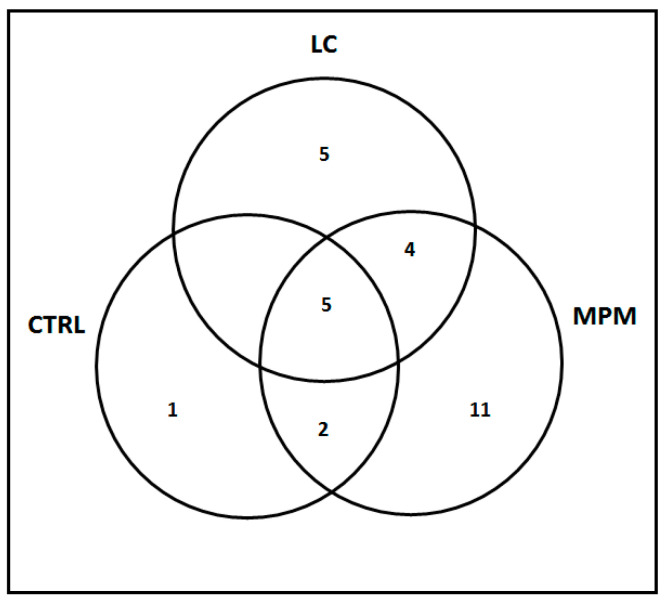
Venn diagram of the exclusive and shared (5) enriched pathways in CTRL (1), MPM (11) and LC (5) populations.

**Figure 2 ijms-23-13628-f002:**
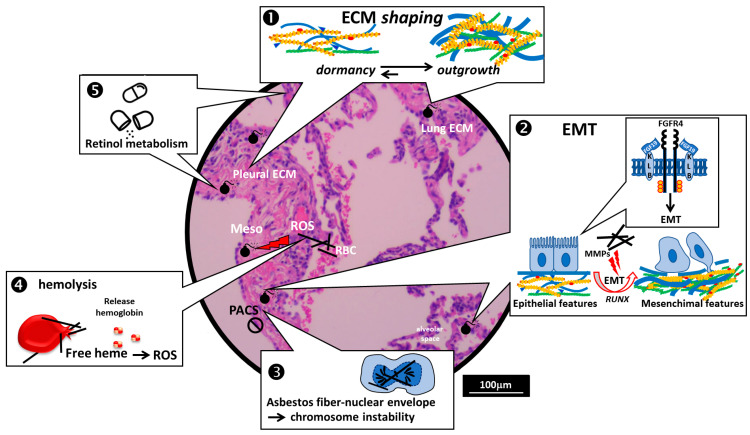
The importance of host genetics in asbestos-induced thoracic cancer susceptibility as deduced from VEA. Protective genetic asset (exclusive to asbestos survivors, indicated with the symbol 
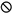
) preserving genomic stability: the possible role of PACS proteins. Susceptibility assets (exclusive to individuals who developed asbestos-related thoracic cancers, indicated with the symbol 
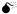
): ➊ECM shaping of the lung/pleural tumor microenvironment in determining dormancy or outgrowth: abnormal ECM dynamics lead to deregulated cell proliferation and invasion, resulting in pathological processes including cancer. ➋ EMT as the initial step in tumorigenesis; ➌ Asbestos fibers and the nuclear envelope. Dysregulation of the nuclear envelope reassembly and susceptibility of developing MPM: a confirmation of the role of the well-documented close interaction between asbestos and the nuclear envelope of mesothelial cells in the induction of chromosome rearrangements seen in MPM; ➍ The emerging role of free heme in MPM pathogenesis: defective scavenging of free heme released by asbestos-fibers-induced hemolysis could contribute to asbestos-induced oxidative stress; ➎ The retinoid (Vitamin A) risk: dysregulated retinoid metabolism in asbestos-exposed individuals could contribute to asbestos-induced thoracic cancer pathogenesis. The histological image is a representative microphotograph relative to a Hematoxylin-and-Eosin-stained section of non-tumor lung which includes both pleural and lung tissue with the sole purpose of acting as a background for the schematic representation. Meso = Mesothelium, Red Blood Cells = RBC, AF = Asbestos Fibers, ROS = Reactive Oxygen Species, ECM = Extracellular Matrix, HS-GAGs = Heparan Sulphate Glycosaminoglycans, HSPGs = Heparan Sulphate ProteoGlycans.

**Table 1 ijms-23-13628-t001:** Clinicopathological characteristics of control, lung cancer and malignant pleural mesothelioma populations.

	CTRL	LC	MPM
Number of cases	5	7	7
Mean age (years ± sd)	79.4 ± 2.2	76.1 ± 8.6	76.9 ± 6.3
Cause of death	Cardiovascular diseases	SCLC [2]; NSCLC [5]	EMPM [2]; BMPM [2]; SMPM [3]
Asbestos bodies count, in n/g dry tissue, (mean ± sd)	1.30 × 10^5^ (6.07 × 10^4^)	1.02 × 10^5^ (1.48 × 10^5^)	1.27 × 10^5^ (9.90 × 10^4^)
Asbestos fibers count, in n/g dry tissue, (mean ± sd)	NA	1.04 × 10^7^ (1.37 × 10^7^)	1.86 × 10^7^ (2.72 × 10^7^)
Number of hyaline plaques			
Absent	0	0	0
Grade 1	0	0	1
Grade 2	1	6	3
Grade 3	4	1	3

CTRL = individuals who did not develop asbestos-related diseases, LC = individuals who developed lung cancer, MPM = individuals who developed malignant pleural mesothelioma, SCLC = small-cell lung cancer, NSCLC = non-small-cell lung cancer, EMPM = epithelial malignant pleural mesothelioma, SMPM = sarcomatoid malignant pleural mesothelioma, BMPM = biphasic malignant pleural mesothelioma, NA = not available.

**Table 2 ijms-23-13628-t002:** Enriched pathways found for the control (CTRL), Lung Cancer (LC) and Malignant Pleural Mesothelioma (MPM) groups.

Group	Reactome ID	Reactome Pathway Name	VariantRatio	BgRatio	OR	CI95−	CI95+	adj. *p*-Value
CTRL	R-HSA-164940	Nef-mediated downregulation of MHC class I complex cell surface expression	12/5610	69/180,208	5.59	2.75	10.40	1.43 × 10^−2^
	R-HSA-2172127	DAP12 interactions	26/5610	311/180,208	2.69	1.73	4.02	4.02 × 10^−2^
	R-HSA-5083632	Defective C1GALT1C1 causes Tn polyagglutination syndrome (TNPS)	148/5610	1809/180,208	2.63	2.20	3.12	1.42 × 10^−19^
	R-HSA-977068	Termination of O-glycan biosynthesis	151/5610	1847/180,208	2.63	2.21	3.11	5.57 × 10^−20^
	R-HSA-5083625	Defective GALNT3 causes familial hyperphosphatemic tumoral calcinosis (HFTC)	148/5610	1811/180,208	2.63	2.20	3.11	1.55 × 10^−19^
	R-HSA-5083636	Defective GALNT12 causes colorectal cancer 1 (CRCS1)	148/5610	1812/180,208	2.62	2.20	3.11	1.61 × 10^−19^
	R-HSA-5621480	Dectin-2 family	151/5610	1892/180,208	2.56	2.15	3.03	4.49 × 10^−19^
	R-HSA-198933	Immunoregulatory interactions between a lymphoid and a non-lymphoid cell	99/5610	1750/180,208	1.82	1.47	2.23	3.00 × 10^−4^
LC	R-HSA-1839128	FGFR4 mutant receptor activation	6/4102	16/180,208	16.48	5.28	44.31	1.63 × 10^−2^
	R-HSA-3656237	Defective EXT2 causes exostoses 2	20/4102	211/180,208	4.16	2.49	6.60	8.21 × 10^−4^
	R-HSA-3656253	Defective EXT1 causes exostoses 1, TRPS2 and CHDS	20/4102	211/180,208	4.16	2.49	6.60	8.21 × 10^−4^
	R-HSA-2024096	HS-GAG degradation	25/4102	312/180,208	3.52	2.24	5.30	6.02 × 10^−4^
	R-HSA-3560801	Defective B3GAT3 causes JDSSDHD	21/4102	270/180,208	3.42	2.08	5.34	8.07 × 10^−3^
	R-HSA-4420332	Defective B3GALT6 causes EDSP2 and SEMDJL1	20/4102	266/180,208	3.30	1.98	5.21	2.17 × 10^−2^
	R-HSA-3560783	Defective B4GALT7 causes EDS, progeroid type	20/4102	269/180,208	3.27	1.96	5.15	2.52 × 10^−2^
	R-HSA-1971475	A tetrasaccharide linker sequence is required for GAG synthesis	21/4102	304/180,208	3.03	1.85	4.73	4.33 × 10^−2^
	R-HSA-975634	Retinoid metabolism and transport	32/4102	571/180,208	2.46	1.67	3.52	2.48 × 10^−2^
	R-HSA-977068	Termination of O-glycan biosynthesis	93/4102	1847/180,208	2.21	1.77	2.73	6.91 × 10^−8^
	R-HSA-5083632	Defective C1GALT1C1 causes Tn polyagglutination syndrome (TNPS)	91/4102	1809/180,208	2.21	1.77	2.74	1.22 × 10^−7^
	R-HSA-5083625	Defective GALNT3 causes familial hyperphosphatemic tumoral calcinosis (HFTC)	91/4102	1811/180,208	2.21	1.76	2.73	1.26 × 10^−7^
	R-HSA-5083636	Defective GALNT12 causes colorectal cancer 1 (CRCS1)	91/4102	1812/180,208	2.21	1.76	2.73	1.29 × 10^−7^
	R-HSA-5621480	Dectin-2 family	92/4102	1892/180,208	2.14	1.71	2.64	5.42 × 10^−7^
MPM	R-HSA-5619063	Defective SLC29A3 causes histiocytosis-lymphadenopathy plus syndrome (HLAS)	6/7745	9/180,208	15.51	4.54	48.84	4.32 × 10^−2^
	R-HSA-1839128	FGFR4 mutant receptor activation	9/7745	16/180,208	13.09	5.10	31.46	9.35 × 10^−4^
	R-HSA-1307965	betaKlotho-mediated ligand binding	9/7745	21/180,208	9.97	4.02	22.72	5.43 × 10^−3^
	R-HSA-2172127	DAP12 interactions	57/7745	311/180,208	4.26	3.15	5.68	3.79 × 10^−14^
	R-HSA-2995410	Nuclear envelope (NE) reassembly	17/7745	93/180,208	4.25	2.38	7.19	6.90 × 10^−3^
	R-HSA-2168880	Scavenging of heme from plasma	17/7745	108/180,208	3.66	2.06	6.14	3.97 × 10^−2^
	R-HSA-3656237	Defective EXT2 causes exostoses 2	26/7745	211/180,208	2.87	1.83	4.32	1.66 × 10^−2^
	R-HSA-3656253	Defective EXT1 causes exostoses 1, TRPS2 and CHDS	26/7745	211/180,208	2.87	1.83	4.32	1.66 × 10^−2^
	R-HSA-8941326	RUNX2 regulates bone development	28/7745	234/180,208	2.78	1.81	4.13	1.25 × 10^−2^
	R-HSA-5663205	Infectious disease	256/7745	2720/180,208	2.19	1.92	2.50	2.76 × 10^−23^
	R-HSA-977068	Termination of O-glycan biosynthesis	170/7745	1847/180,208	2.14	1.82	2.51	3.42 × 10^−14^
	R-HSA-5083632	Defective C1GALT1C1 causes Tn polyagglutination syndrome (TNPS)	166/7745	1809/180,208	2.14	1.81	2.51	1.12 × 10^−13^
	R-HSA-5083625	Defective GALNT3 causes familial hyperphosphatemic tumoral calcinosis (HFTC)	166/7745	1811/180,208	2.13	1.81	2.51	1.17 × 10^−13^
	R-HSA-5083636	Defective GALNT12 causes colorectal cancer 1 (CRCS1)	166/7745	1812/180,208	2.13	1.80	2.50	1.20 × 10^−13^
	R-HSA-975634	Retinoid metabolism and transport	51/7745	571/180,208	2.08	1.53	2.77	1.56 × 10^−2^
	R-HSA-5621480	Dectin-2 family	168/7745	1892/180,208	2.07	1.75	2.42	9.39 × 10^−13^
	R-HSA-3000157	Laminin interactions	79/7745	988/180,208	1.86	1.46	2.34	2.36 × 10^−3^
	R-HSA-909733	Interferon alpha/beta signaling	79/7745	1010/180,208	1.82	1.43	2.29	6.34 × 10^−3^
	R-HSA-198933	Immunoregulatory interactions between a lymphoid and a non-lymphoid cell	131/7745	1750/180,208	1.74	1.45	2.08	3.45 × 10^−5^
	R-HSA-3000171	Non-integrin membrane–ECM interactions	108/7745	1556/180,208	1.61	1.31	1.97	1.86 × 10^−2^
	R-HSA-6805567	Keratinization	108/7745	1563/180,208	1.61	1.31	1.96	1.97 × 10^−2^
	R-HSA-1474228	Degradation of the extracellular matrix	185/7745	2814/180,208	1.53	1.31	1.78	4.04 × 10^−4^

The information featured in the table is: Reactome ID; Reactome pathway name; the ratio of group common variant in the pathway (shared in each group of patients) per group total common variant found in all patients (VariantRatio); the ratio of group common variant in the pathway per group total common variant in the reference dataset (BgRatio); odds ratio (OR); 95% lower and up confidence intervals (CI95− and CI95+); and FDR-adjusted *p*-values of the Fisher’s exact test (adj. *p*-value).

## Data Availability

Not applicable.

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
