# Peer review of "Variant Enrichment Analysis to Explore Pathways Disruption in a Necropsy Series of Asbestos-Exposed Shipyard Workers"

_ijms, 2022, doi:10.3390/ijms232113628_

Round 1

Reviewer 1 Report

The manuscript is based on published variants analysis pipeline to identify possible disrupted pathways in individuals who developed lung cancer or mesothelioma

  1. Formalin produces artificial C > T or G > A mutations, Can the authors describe how to remove these artificial mutations in the method?
  2. Line 135-138, the authors were not able to any variants described by GWAS, does that mean the VEA pipeline would miss many risk variants that could be found by other genetic analysis methods, such as GWAS? Could the authors find any genes enriched by both GWAS and the VEA pipeline?
  3. In Figure 1, five enriched pathways were shared by control samples and patient samples, would that suggest that many non-disease related pathways could also be enriched by the VEA pipeline when investigating the patient samples?

Author Response

Reviewer 1

  1. Formalin produces artificial C > T or G > A mutations, Can the authors describe how to remove these artificial mutations in the method?

We were aware of possible artificial variations in the DNA extracted by FFPE. In the methods section we described the comparison of the two DNA extraction methods that led to the choice of the Promega DNA extraction kit. We’ll evidence this artifact issue in the methods section (lines 460-465). We also considered the removal of duplicated and soft-clipped reads, as well as filter out low coverage variants. We added further details in the methodology section (lines 498-507).

Despite the possibility of artifacts, reduced by our quality control pipeline, the quality of our DNA samples was good enough as per the recommendations of our NGS service provider Macrogen.

Line 135-138, the authors were not able to any variants described by GWAS, does that mean the VEA pipeline would miss many risk variants that could be found by other genetic analysis methods, such as GWAS? Could the authors find any genes enriched by both GWAS and the VEA pipeline?

The MPM risk variants already found in GWAS have not been excluded at all but considered in our VEA pipeline. However, we have not found any of the variants previously associated with asbestos exposure (i.e. rs13383928 e rs9635542 [https://www.ebi.ac.uk/gwas/efotraits/EFO_0004806) in our data set. Moreover, we want to reinforce the concept that the VEA cannot be compared to GWAS since it has not been designed to highlight variations enriched genes, but it is aimed at identifying enriched pathways based on genetic variants analysis using a bioinformatic pipeline already described and recalled in the methods section of our manuscript. Finally, VEA has been designed to work in datasets characterized by small number of individuals as in our study.

  1. In Figure 1, five enriched pathways were shared by control samples and patient samples, would that suggest that many non-disease related pathways could also be enriched by the VEA pipeline when investigating the patient samples?

We do thank the reviewer for the valuable comment. The answer is Yes; VEA is a pipeline that discover pathways based on the genetic variations found by WES (or Whole genome sequencing), according to the pipeline described in the methods section. Our VEA approach was not based on cases-controls comparison, but just aimed at evidencing enriched pathways, both exclusive or in common within the three groups analyzed in this study. Now, asbestos exposure function as exposome and the response to its exposure will depend upon the genomic background of the individuals analyzed. So, we can expect shared pathways between different phenotypes, in our case having in common the asbestos exposure, as per the 5 common pathways shared by MPM, LC and CTRL. Notably, despite having the same pattern of asbestos exposure and some shared pathways, MPM, LC and CTRL showed specific enriched pathways, each one characteristic of each group, so with the same asbestos exposure, group specific genetic variants contributing to VEA results, could account for the three different phenotypes observed. In patients with LC and MPM the Exclusive enriched pathways were unique with respect to the individual of the other groups as well as to the GNOMAD 3.0 genome, used as general control.

Reviewer 2

I reviewed the manuscript ijms-1851044 by Crovella et al. The authors examined autopsy specimens from asbestos-exposed individuals, who either developed benign pleural plaques, malignant mesothelioma or lung cancer (both SCLC and NSCLC). They performed variant enrichment analysis in those specimens and they identified disrupted gene pathways in patients with asbestos-linked thoracic malignancies and pathways with possible protective function against neoplasms in patients with asbestos-related pleural disease. The most important issues are the following:

1) The study population is too small and highly heterogeneous. This undermines the study findings.

We cordially disagree with the reviewer’s statement “highly heterogeneous”. In fact, we are aware that the main limitation of our genetic study is represented by the low number of enrolled cases, due to the extremely restrictive sample inclusion criteria of the study subjects. This selection was the result of the choice:

  • to create an “asbestos-resistant exposed population” sample which could be suitably compared with the MPM/LC group. Exposed subjects with the highest MPM/LC-free follow-up were selected as controls in this study. Inclusion criteria: objective signs confirming asbestos exposure who didn’t develop pleural or peritoneal malignant mesothelioma, lung carcinoma or lung asbestosis, and died of other causes after the age of 75. The 75 years cut-off in the control population was chosen on the basis of previously reported data on latency periods (defined as the time interval between first exposure to asbestos and death), types of occupational exposure and age at time of death, obtained from a series of MPM from the same geographic area (Bianchi et al. 2007 in the References section).
  • to obtain a highly homogenous (rather than the highly heterogeneous) population as the exposure criteria are concerned.

Finally, being quite low the number of individuals enrolled for this study after WES analysis we used the Variant Enrichment Approach already known to be suitable for low number samples (see the recent article doi: 10.3390/ijms2304227). We mentioned this issue in the paragraph 3.2. Strength and limitation of the study (lines 409-414).

2) The smoking history of the study participants is not presented in control and MPM patients.

We do thank the reviewer for the valuable suggestion. We did not report smoking history in MPM patients since, up to now, no association between cigarette smoking and mesothelioma has been observed [Muscat, J. E., & Wynder, E. L. (1991). Cigarette smoking, asbestos exposure, and malignant mesothelioma. Cancer research, 51(9), 2263–2267]; moreover this information is not usually reported in clinical documents of mesothelioma patients, while, since the interaction between asbestos and smoking is multiplicative of the risk of lung cancer  (Markowitz et al. 2013, https://doi.org/10.1164/rccm.201302-0257OC), this information is often reported in clinical history of asbestos exposed lung cancer patients. Aware of this limitation, we have repeatedly reiterated it in the text that LC population is a population of asbestos exposed smoking shipyards (lines 111 and 449).

Furthermore all patients in LC group were smokers. It is well established in literature that non-smoking patients with lung cancer have distinct genetic profile compared to smokers.

A different lung cancer genetic profile is well established between never and ever smokers (eg. Chapman et al 2016. https://doi.org/10.1016), anyway inherited susceptibility to lung cancer risk in never smokers is still poorly understood and, as recently reported by Hung and colleagues (Hung et al 2019. DOI: 10.1016/j.jtho.2019.04.008), is associated to genetic variants with pan-cancer risk effects. Being the scope of this study the comparison of genetic factors impacting different molecular pathways in asbestos exposed shipyard workers, the effect of smoking on LC population can be considered multiplicative the risk of cancer. Finally, in the exposome factors involved in the susceptibility to mesothelioma, asbestos exposure is widely accepted as a risk factor, while smoking is still to be unraveled and looks like to have a different weight on the susceptibility to the disease.

3) The authors should explain why they did not evaluate biopsy specimens from living individuals as well.

The present study has been designed aimed at analyzing germinal DNA, so focusing on the genetic background of asbestos exposed individuals, but not tumor DNA obtainable from biopsy specimens. Genomic was extracted from formalin-fixed and paraffin-embedded (FFPE) archived autopsy heart tissue slices as previously described (Borelli et al. 2015 in the Reference section). The myocardial tissue of all autopsy cases was always free of neoplastic cells and was chosen precisely to avoid bias due to somatic alterations in tumor tissue, such as loss of heterozygosity.

Furthermore, the strength of this study is represented precisely by the availability of sample that are proven by autopsy to be from patients exposed to asbestos with or without mesothelioma or lung cancer rather than the usual collection of cases based on “history of exposure” that often are unreliable. In this study we selected a highly homogenous population as objective exposure criteria are concerned: evaluation of lung asbestos bodies and fibers content and presence of pleural plaques, together with the same occupational history.  …as reported in the manuscript:

  • Introduction section - A population-based autopsy study was designed in which asbestos exposure, whose assessment represented a relevant source of heterogeneity in previous genetic studies (Ugolini et al, 2008; Betti et al, 2018) (13, 21), was evaluated and quantitated by investigating objective signs of exposure (Bianchi et al, 2001) (33).
  • Result section - All individuals were employed in the shipyard and displayed similar asbestos exposure due to working condition, which was validated by the presence of pleural plaques and characterized qualitatively (type of asbestos) and quantitively by the analysis of asbestos (bodies and/or fibers: ABs and AFs, respectively) lung burden. Pleural plaques of grade 2 were mostly present in all cases, except one of grade 1 in the MPM group.

4) The authors examined a group of people from a small region in Italy. This is making difficult to generalize the study findings. However, this population may exhibit a specific genetic profile. In order to unveil whether there is a familial predisposal for the development of thoracic neoplasms upon asbestos exposure in this specific population extra control groups should be added in the evaluation with VEA. (E.g. healthy individuals and patients with lung cancer but with no asbestos exposure).

We appreciated the reviewer’s criticism. Firstly, we were well-aware that the study population was from an area of Friuli Venezia Giulia region where the shipyards are installed. In the patients and controls selection we carefully searched for any parental relationship between them, in fact familiarity or consanguinity were exclusion criteria for a genetic study. Secondly, the choice of the so called “small region” is related to the shipyards, here localized and source of asbestos. In the VEA the results are routinely compared with the source population (in this case the Non-Finnish European controls as per Gnomad 3.0). Third, the scope of this article was the comparison of genetic factors impacting different molecular pathways in asbestos exposed shipyard workers. The comparison with individual with lung cancer, but no asbestos exposure was not making sense in our study, that is not an association study requiring high number of individuals. We voluntarily restricted the sample study by comparing a homogeneous group of individuals.

See also our previous studies:

Borelli et al, 2015. https://doi.org/10.1186/s13027-015-0022-0

Crovella et al, 2016. https://doi.org/10.1080/15287394.2015.1123452

Crovella et al, 2018. https://doi.org/10.1080/15287394.2017.1416911

Celsi et al, 2019. https://doi.org/10.1080/15287394.2019.1694612

5) The study hypothesis is not clearly depicted. The description of statistical analysis is lacking. In the introduction section the authors should emphasize the advantages of VEA versus GWAS analysis.

We thank the reviewer for the valuable comment, and we try to better elucidate the VEA.

It should be clear for the reader that VEA is not designed to overcome/substitute GWAS, both could be used. However, GWAS methods require large number of individuals, in all groups, to reach variants with small impact on phenotype. Moreover, the GWAS is focused on a genetic variant-based view. VEA has a different approach and perspective, which is pathway enrichment analysis based on genetic variants. So, the objective of VEA is to identify Exclusive Enriched Pathways specific for any of the studied groups, even when the number of enrolled individuals is low. We adjusted the text, accordingly, see Lines: 78-81

6) The discussion is too large and in certain points is out of scope of the manuscript.

The length of the discussion responds to the need of discuss the relevance of the pathway highlighted by VEA on the basis of the existing literature. Anyway, in agreement with the reviewer we have shortened it to make it more fluent for the readers.

7) The authors jump into conclusions not supported by the study findings.

We acknowledge the criticism, but we cannot understand what the reviewer wants to highlight: the conclusion is based on Exclusive Enriched Pathways specific of each study group and in the conclusions, we simply emphasize that the use of VEA can allow the analysis of such a small group of individuals.

8) The manuscript has got an odd structure, is hard to follow and it needs grammatical and vocabulary editing.

We enrolled a mother tongue scientist to correct the grammar and the structure of the manuscript to make it more understandable for the readers.

Reviewer 2 Report

I reviewed the manuscript ijms-1851044 by Crovella et al. The authors examined autopsy specimens from asbestos-exposed individuals, who either developed benign pleural plaques, malignant mesothelioma or lung cancer (both SCLC and NSCLC). They performed variant enrichment analysis in those specimens and they identified disrupted gene pathways in patients with asbestos-linked thoracic malignancies and pathways with possible protective function against neoplasms in patients with asbestos-related pleural disease. The most important issues are the following:

-The study population is too small and highly heterogeneous. This undermines the study findings.

-The smoking history of the study participants is not presented in control and MPM patients. Furthermore all patients in LC group were smokers. It is well established in literature that non-smoking patients with lung cancer have distinct genetic profile compared to smokers.

-The authors should explain why they did not evaluate biopsy specimens from living individuals as well.   

-The authors examined a group of people from a small region in Italy. This is making difficult to generalize the study findings. However, this population may exhibit a specific genetic profile. In order to unveil whether there is a familial predisposal for the development of thoracic neoplasms upon asbestos exposure in this specific population extra control groups should be added in the evaluation with VEA. (E.g. healthy individuals and patients with lung cancer but with no asbestos exposure).

-The study hypothesis is not clearly depicted. The description of statistical analysis is lacking. In the introduction section the authors should emphasize  the advantages of VEA versus GWAS analysis. The discussion is too large and in certain points is out of scope of the manuscript. The authors jump into conclusions not supported by the study findings.

-The manuscript has got an odd structure, is hard to follow and it needs grammatical and vocabulary editing.  

Author Response

Reviewer 2

I reviewed the manuscript ijms-1851044 by Crovella et al. The authors examined autopsy specimens from asbestos-exposed individuals, who either developed benign pleural plaques, malignant mesothelioma or lung cancer (both SCLC and NSCLC). They performed variant enrichment analysis in those specimens and they identified disrupted gene pathways in patients with asbestos-linked thoracic malignancies and pathways with possible protective function against neoplasms in patients with asbestos-related pleural disease. The most important issues are the following:

1) The study population is too small and highly heterogeneous. This undermines the study findings.

We cordially disagree with the reviewer’s statement “highly heterogeneous”. In fact, we are aware that the main limitation of our genetic study is represented by the low number of enrolled cases, due to the extremely restrictive sample inclusion criteria of the study subjects. This selection was the result of the choice:

  • to create an “asbestos-resistant exposed population” sample which could be suitably compared with the MPM/LC group. Exposed subjects with the highest MPM/LC-free follow-up were selected as controls in this study. Inclusion criteria: objective signs confirming asbestos exposure who didn’t develop pleural or peritoneal malignant mesothelioma, lung carcinoma or lung asbestosis, and died of other causes after the age of 75. The 75 years cut-off in the control population was chosen on the basis of previously reported data on latency periods (defined as the time interval between first exposure to asbestos and death), types of occupational exposure and age at time of death, obtained from a series of MPM from the same geographic area (Bianchi et al. 2007 in the References section).
  • to obtain a highly homogenous (rather than the highly heterogeneous) population as the exposure criteria are concerned.

Finally, being quite low the number of individuals enrolled for this study after WES analysis we used the Variant Enrichment Approach already known to be suitable for low number samples (see the recent article doi: 10.3390/ijms2304227). We mentioned this issue in the paragraph 3.2. Strength and limitation of the study (lines 409-414).

2) The smoking history of the study participants is not presented in control and MPM patients.

We do thank the reviewer for the valuable suggestion. We did not report smoking history in MPM patients since, up to now, no association between cigarette smoking and mesothelioma has been observed [Muscat, J. E., & Wynder, E. L. (1991). Cigarette smoking, asbestos exposure, and malignant mesothelioma. Cancer research, 51(9), 2263–2267]; moreover this information is not usually reported in clinical documents of mesothelioma patients, while, since the interaction between asbestos and smoking is multiplicative of the risk of lung cancer  (Markowitz et al. 2013, https://doi.org/10.1164/rccm.201302-0257OC), this information is often reported in clinical history of asbestos exposed lung cancer patients. Aware of this limitation, we have repeatedly reiterated it in the text that LC population is a population of asbestos exposed smoking shipyards (lines 111 and 449).

Furthermore all patients in LC group were smokers. It is well established in literature that non-smoking patients with lung cancer have distinct genetic profile compared to smokers.

A different lung cancer genetic profile is well established between never and ever smokers (eg. Chapman et al 2016. https://doi.org/10.1016), anyway inherited susceptibility to lung cancer risk in never smokers is still poorly understood and, as recently reported by Hung and colleagues (Hung et al 2019. DOI: 10.1016/j.jtho.2019.04.008), is associated to genetic variants with pan-cancer risk effects. Being the scope of this study the comparison of genetic factors impacting different molecular pathways in asbestos exposed shipyard workers, the effect of smoking on LC population can be considered multiplicative the risk of cancer. Finally, in the exposome factors involved in the susceptibility to mesothelioma, asbestos exposure is widely accepted as a risk factor, while smoking is still to be unraveled and looks like to have a different weight on the susceptibility to the disease.

3) The authors should explain why they did not evaluate biopsy specimens from living individuals as well.

The present study has been designed aimed at analyzing germinal DNA, so focusing on the genetic background of asbestos exposed individuals, but not tumor DNA obtainable from biopsy specimens. Genomic was extracted from formalin-fixed and paraffin-embedded (FFPE) archived autopsy heart tissue slices as previously described (Borelli et al. 2015 in the Reference section). The myocardial tissue of all autopsy cases was always free of neoplastic cells and was chosen precisely to avoid bias due to somatic alterations in tumor tissue, such as loss of heterozygosity.

Furthermore, the strength of this study is represented precisely by the availability of sample that are proven by autopsy to be from patients exposed to asbestos with or without mesothelioma or lung cancer rather than the usual collection of cases based on “history of exposure” that often are unreliable. In this study we selected a highly homogenous population as objective exposure criteria are concerned: evaluation of lung asbestos bodies and fibers content and presence of pleural plaques, together with the same occupational history.  …as reported in the manuscript:

  • Introduction section - A population-based autopsy study was designed in which asbestos exposure, whose assessment represented a relevant source of heterogeneity in previous genetic studies (Ugolini et al, 2008; Betti et al, 2018) (13, 21), was evaluated and quantitated by investigating objective signs of exposure (Bianchi et al, 2001) (33).
  • Result section - All individuals were employed in the shipyard and displayed similar asbestos exposure due to working condition, which was validated by the presence of pleural plaques and characterized qualitatively (type of asbestos) and quantitively by the analysis of asbestos (bodies and/or fibers: ABs and AFs, respectively) lung burden. Pleural plaques of grade 2 were mostly present in all cases, except one of grade 1 in the MPM group.

4) The authors examined a group of people from a small region in Italy. This is making difficult to generalize the study findings. However, this population may exhibit a specific genetic profile. In order to unveil whether there is a familial predisposal for the development of thoracic neoplasms upon asbestos exposure in this specific population extra control groups should be added in the evaluation with VEA. (E.g. healthy individuals and patients with lung cancer but with no asbestos exposure).

We appreciated the reviewer’s criticism. Firstly, we were well-aware that the study population was from an area of Friuli Venezia Giulia region where the shipyards are installed. In the patients and controls selection we carefully searched for any parental relationship between them, in fact familiarity or consanguinity were exclusion criteria for a genetic study. Secondly, the choice of the so called “small region” is related to the shipyards, here localized and source of asbestos. In the VEA the results are routinely compared with the source population (in this case the Non-Finnish European controls as per Gnomad 3.0). Third, the scope of this article was the comparison of genetic factors impacting different molecular pathways in asbestos exposed shipyard workers. The comparison with individual with lung cancer, but no asbestos exposure was not making sense in our study, that is not an association study requiring high number of individuals. We voluntarily restricted the sample study by comparing a homogeneous group of individuals.

See also our previous studies:

Borelli et al, 2015. https://doi.org/10.1186/s13027-015-0022-0

Crovella et al, 2016. https://doi.org/10.1080/15287394.2015.1123452

Crovella et al, 2018. https://doi.org/10.1080/15287394.2017.1416911

Celsi et al, 2019. https://doi.org/10.1080/15287394.2019.1694612

5) The study hypothesis is not clearly depicted. The description of statistical analysis is lacking. In the introduction section the authors should emphasize the advantages of VEA versus GWAS analysis.

We thank the reviewer for the valuable comment, and we try to better elucidate the VEA.

It should be clear for the reader that VEA is not designed to overcome/substitute GWAS, both could be used. However, GWAS methods require large number of individuals, in all groups, to reach variants with small impact on phenotype. Moreover, the GWAS is focused on a genetic variant-based view. VEA has a different approach and perspective, which is pathway enrichment analysis based on genetic variants. So, the objective of VEA is to identify Exclusive Enriched Pathways specific for any of the studied groups, even when the number of enrolled individuals is low. We adjusted the text, accordingly, see Lines: 78-81

6) The discussion is too large and in certain points is out of scope of the manuscript.

The length of the discussion responds to the need of discuss the relevance of the pathway highlighted by VEA on the basis of the existing literature. Anyway, in agreement with the reviewer we have shortened it to make it more fluent for the readers.

7) The authors jump into conclusions not supported by the study findings.

We acknowledge the criticism, but we cannot understand what the reviewer wants to highlight: the conclusion is based on Exclusive Enriched Pathways specific of each study group and in the conclusions, we simply emphasize that the use of VEA can allow the analysis of such a small group of individuals.

8) The manuscript has got an odd structure, is hard to follow and it needs grammatical and vocabulary editing.

We enrolled a mother tongue scientist to correct the grammar and the structure of the manuscript to make it more understandable for the readers.

Reviewer 3 Report

The authors described using variant enrichment analysis to investigate potentially altered pathways in cases of asbestos exposure leading to lung cancer or mesothelioma.  The research question is interesting and the use of whole exome sequencing for variant enriched pathways is appropriate, however, the restrictions on sample inclusion limit any conclusions that may be drawn.  

I have the following questions and comments:

General:  There was a great deal of highlighted text in the document that I received.  This was distracting.

pg. 1, line 2:  There is inconsistent capitalization in the title.

pg. 1, line 32:  The abstract contains no statement of conclusions.

pg. 2, lines 94-97:  The last paragraph in the introduction is redundant and could be deleted.

pg. 3, line 109:  Smoking history is important with regard to pulmonary disease risk.  Furthermore, current and former smoking history may further divide the LC group, depending on pack years.   

pg. 3, line 121:  Should the threshold be <1 million/g?

pg. 4, lines 139-140: The strength of this conclusion is limited due to the small number of subjects in the study.  The results of the current study is inconsistent with simple genetic variants controlling multifactorial genotypes.  Is this belief based on lack of previous reports on association of genetic variants in asbestos-related cancers?  Although no single mutations from the pathways were reported elsewhere, were there mutations outside of the pathways found in the current study?

pg. 7, line 199: What is meant by “generic”? Does the combination of these pathways in the asbestos related samples not suggest a new role for these pathways? Is the one enriched pathway in the CTRL group considered generic?

pg. 7, lines 212-214: Were the samples collected as early as 1985?  Was there any observation of tumor mutation burden vs date of sample fixation?

pg. 11, lines 444-446: How were the seven LC cases subdivided into these categories?  Were there observations related to the particular classifications?

pg. 13, line 456: Was tumor tissue selected from the slides by micro-dissection? If not, what was percent tumor in the sections?

pg. 14, line 504: Does AD refer to allele frequency (percent of reads) or coverage? A coverage limit of 10 seems low.

Tables

Table 1.  Explanation of the headings in the table should be in a footnote, rather than in the title. 

Table 2.  Explanation of the headings in the table should be in a footnote, rather than in the title.  There isn’t a column for the Fisher’s exact test.

pg. 4, line 185: Please define “group total common variant”.

Figures

Figure 1.  Should “MEM” in the image be MPM? “MEM” is also indicated in Table 2.

Figure 2.  The stained section diagram is difficult to relate with the pathways depicted.  Perhaps a higher magnification image with clearer depiction of cellular and extracellular pleural and lung tissue would clarify the intent of the figure. 

Supplementary files

The sample categories (CTRL, MPM, LC) are not indicated in the supplementary files.

Author Response

Reviewer 3

Comments and Suggestions for Authors

We thank the reviewer for the accurate and detailed overview. By taking into account his/her comments we think that, besides answering to all their concerns, we managed to improve and strengthen the whole work. Below you can find the answers of the authors point by point. Relative changes in the manuscript text are highlighted in green.

  1. 1, line 2: There is inconsistent capitalization in the title.

Authors’ answer.

The reviewer is right, we apologize for the mistake and we have changed the text accordingly: “Variant Enrichment Analysis to explore pathways disruption in a necropsy series of asbestos-exposed shipyard workers”.

  1. 1, line 32: The abstract contains no statement of conclusions.

Authors’ answer.

We thank the reviewer for the suggestion, based on which we added the following paragraph:

“By using VEA analysis we confirmed the involvement of pathways considered as the main responsible for asbestos induced carcinogenesis: oxidative stress and chromosome instability. Furthermore, we identified protective genetic assets preserving genome stability and susceptibility assets predisposing to a worst outcome”.

  1. 2, lines 94-97: The last paragraph in the introduction is redundant and could be deleted.

Authors’ answer.

In agreement with the reviewer's suggestion we have deleted the last paragraph: “Formalin fixed paraffin embedded (FFPE) tissues were used as source of DNA and WES analysis was performed. VEA workflow was employed to identify potentially disrupted pathways in individuals who developed thoracic cancers induced by comparable and objectively established asbestos exposure”.

  1. 3, line 109: Smoking history is important with regard to pulmonary disease risk. Furthermore, current and former smoking history may further divide the LC group, depending on pack years.  

Authors’ answer.

We agree with the reviewer comment, but unfortunately this datum is not available.

  1. 3, line 121: Should the threshold be <1 million/g?

Authors’ answer.

The correct form in that in the text (>1 million/g) since for clinical purposes, the following guidelines are recommended to identify persons with a high probability of exposure to asbestos dust at work: “over 0.1 milllion amphibole fibers (>5 μm) per gram of dry lung tissue or over 1 million amphibole fibers (>1 μm) per gram of dry lung tissue”, as reported in reference 35.

  1. 4, lines 139-140: The strength of this conclusion is limited due to the small number of subjects in the study. The results of the current study is inconsistent with simple genetic variants controlling multifactorial genotypes. Is this belief based on lack of previous reports on association of genetic variants in asbestos-related cancers? Although no single mutations from the pathways were reported elsewhere, were there mutations outside of the pathways found in the current study?

Authors’ answer.

We are aware that the main limitation of our genetic study is represented by the low number of enrolled cases, due to the extremely restrictive sample inclusion criteria of the study subjects. This selection was the result of the choice:

  • to create an “asbestos-resistant exposed population” sample which could be suitably compared with the MPM/LC group. Exposed subjects with the highest MPM/LC-free follow-up were selected as controls in this study. Inclusion criteria: objective signs confirming asbestos exposure who didn’t develop pleural or peritoneal malignant mesothelioma, lung carcinoma or lung asbestosis, and died of other causes after the age of 75.
  • to obtain a highly homogenous population as the exposure criteria are concerned.

Finally, being quite low the number of individuals enrolled for this study after WES analysis we used the recently developed Variant Enrichment Approach already known to be suitable for low number samples (see our recent article doi: 10.3390/ijms23042278).

The MPM risk variants already found in GWAS have not been excluded at all but considered in our VEA pipeline. However, we have not found any of the variants previously associated with asbestos exposure (i.e. rs13383928 e rs9635542 [https://www.ebi.ac.uk/gwas/efotraits/EFO_0004806) in our data set. Moreover, we want to reinforce the concept that the VEA cannot be compared to GWAS since it has not been designed to highlight variations enriched genes, but it is aimed at identifying enriched pathways based on genetic variants analysis using a bioinformatic pipeline already described and recalled in the methods section of our manuscript.

  1. 7, line 199: What is meant by “generic”? Does the combination of these pathways in the asbestos related samples not suggest a new role for these pathways? Is the one enriched pathway in the CTRL group considered generic?

Authors’ answer.

We do thank the reviewer for having raised this point. The term “generic” is inappropriate. The pathways called “generic” were the ones not involved in neoplastic transformation and common to most of the activities performed by a normal cell. So, we changed in the text of the manuscript the generic misleading term, specifying that the pathways are not involved in the neoplastic transformation: “These pathways are not involved in the neoplastic transformation and, as they do not contribute to the determination of the susceptibility to develop asbestos-related cancers, they will not be further considered in this manuscript”.

  1. 7, lines 212-214: Were the samples collected as early as 1985? Was there any observation of tumor mutation burden vs date of sample fixation?

Authors’ answer.

In this study we didn’t perform genetic analysis on tumor DNA, but on germinal DNA extracted form myocardial tissue. The myocardial tissue of all autopsy cases was always free of neoplastic cells and was chosen to avoid bias due to somatic alterations in tumor tissue, such as loss of heterozygosity.

  1. 11, lines 444-446: How were the seven LC cases subdivided into these categories? Were there observations related to the particular classifications?

Authors’ answer.

The reviewer is right, we apologize for the mistake and we have changed the text accordingly: “Most Lung cancers were classified into two large histological categories: small cell lung cancer (SCLC n=2) and or non-small cell lung cancer (N=7)., further subdivided into adenocarcinoma, 445 squamous carcinoma, and large cell carcinoma (97)”.

  1. 13, line 456: Was tumor tissue selected from the slides by micro-dissection? If not, what was percent tumor in the sections?

Authors’ answer.

As reported in the Material and Methods section “Myocardial tissue was chosen  as the starting material for (germinal an not tumor) DNA extraction, being free from neoplastic cells and thus without somatic alterations due to tumorigenic transformation (98)”.

Slices were obtained directly from the paraffin block: 40–50 slices were cut with a 5–7 μm thickness and processed for DNA extraction.

  1. 14, line 504: Does AD refer to allele frequency (percent of reads) or coverage? A coverage limit

Authors’ answer.

AD stands for “Allele Depth”, which is the sum of reads covering this locus

Tables

Table 1.  Explanation of the headings in the table should be in a footnote, rather than in the title. 

Authors’ answer.

We thank the reviewer for the suggestion, based on which we made the requested change.

Table 2.  Explanation of the headings in the table should be in a footnote, rather than in the title.  There isn’t a column for the Fisher’s exact test.

Authors’ answer.

Indeed, there was an error with the heading. We thank the reviewer for the suggestion, based on which we made the requested change.

  1. 4, line 185: Please define “group total common variant”.

Authors’ answer.

The term “group total common variant” has been amended and explained in the text (footnote in table 2) as follows: “The ratio of group common variant in the pathway (shared in each group of patients) per group total common variant found in all patients (VariantRatio)”.

Figures

Figure 1.  Should “MEM” in the image be MPM? “MEM” is also indicated in Table 2.

Authors’ answer.

We apologize for the typo we have corrected.

Figure 2.  The stained section diagram is difficult to relate with the pathways depicted.  Perhaps a higher magnification image with clearer depiction of cellular and extracellular pleural and lung tissue would clarify the intent of the figure. 

Authors’ answer.

We agree with the reviewer, however the histological image has the sole purpose of acting as a background for the schematic representation and  represents the best compromise between the constraints imposed on the size of the images and the completeness of the information contained. The stained section diagram was the best that we have been able to obtain.

Supplementary files

The sample categories (CTRL, MPM, LC) are not indicated in the supplementary files.

Authors’ answer.

We apologize for the omission, group information has now been added accordingly.

Round 2

Reviewer 1 Report

I have no further comments

Author Response

We thank thWe thank the reviewer for the accurate and detailed overview.